# Synthetic Conditions, Physical Properties, and Antibacterial Activities of Silver Nanoparticles with Exopolysaccharides of a Medicinal Fungus

**DOI:** 10.3390/ma15165620

**Published:** 2022-08-16

**Authors:** Xingyun Yang, Jian-Yong Wu

**Affiliations:** Department of Applied Biology & Chemical Technology, The Hong Kong Polytechnic University, Hung Hom, Kowloon, Hong Kong

**Keywords:** silver nanoparticle, fungal polysaccharide, reaction conditions, antibacterial activity

## Abstract

Natural polysaccharides are attractive and promising biomacromolecules for the green synthesis of silver nanoparticles (Ag NPs) with a broad spectrum of useful functions. This study aims to evaluate the synthetic conditions and physical properties of Ag NPs using three fractions of exopolysaccharide (EPS), namely EPS-1, EPS-2, and EPS-3, produced by a medicinal fungus known as Cs-HK1, with variations in their chemical composition and molecular weight. Each of the EPS fractions had a unique set of optimal synthetic conditions (reaction time course, temperature, and reagent concentration), resulting in a specific range of Ag NP size distributions. The Ag NPs synthesized with the EPS-1 fraction had the smallest particle size (~160 nm) and the most significant antibacterial activities against *Escherichia coli* (Gram−) and *Staphylococcus aureus* (Gram+), with a minimal inhibitory concentration (MIC) of 0.2 mg/mL on *E. coli* and 0.075 mg/mL on *S. aureus*. The results proved the success of the scheme of this green synthesis scheme with all three EPS fractions and the potential antibacterial application of EPS-coated Ag NPs.

## 1. Introduction

Silver nanoparticles (Ag NPs) have attracted vast research interest because of their potential functions in many fields such as biomedical, environmental, electronic, catalysis, and antimicrobial areas [1,2,3,4,5,6]. Compared with other metallic NPs, Ag NPs exhibit outstanding antibacterial activity, which is needed for many medical and biological applications [7]. The antibacterial property of Ag NPs is particularly useful for surface-coating agents and wound dressings [8]. As shown by a previous study [9,10], Ag NPs can be coated on textile fabrics, particularly on the sheath part, with more significant antibacterial properties than when coated on the core part.

Chemical reduction is one of the most popular bottom-up approaches for the preparation of Ag NPs as a colloidal dispersion in water or organic solvents [11,12,13,14]. Conversion of Ag(I) ions to atomic Ag (0) can be achieved with reducing agents such as NaBH_4_ [15,16], citrate [17], and elemental hydrogen [18]. Ag NPs formed with strong reductants such as borohydride are extremely small [19], and weaker reductants such as citrate would result in a broad particle size distribution at a lower reaction rate [20]. In addition to the reducing agents, capping agents need to be added to the reaction mixture as stabilizers, which bind the particles in order to prevent aggregation [21]. Polymers such as polyvinyl pyrolidine (PVP) [22] and bovine serum albumin (BSA) [2], and organic molecules such as polysaccharides and polyphenol usually serve as the stabilizers [23].

A major concern with the use of chemical reducing and capping agents is their potential toxicity, which is unfavorable for applications related to humans, such as in food, cosmetics, and pharmaceutical products. Natural and biological molecules such as polysaccharides have been widely explored as biocompatible agents for the green synthesis of Ag NPs with low toxicity [24,25]. It has been suggested that the aldehyde groups in polysaccharide molecules may contribute to the Ag+ ion reduction [26,27]. As some of these polysaccharides may act as both reducing and capping agents, synthesis and stabilization can be accomplished in a single step.

*Cordyceps sinensis* is a valuable medicinal fungus and has been used as a tonic food in China since ancient times. Cs-HK1 is an anamorphic fungus isolated from the fruit body of wild *C. sinensis* [28]. In a previous research study of our group, we established a novel green method to synthesize uniformly sized Ag NPs using exopolysaccharides (EPS) produced by the Cs-HK1 fungus [27]. In this study, we further investigated the conditions for Ag NP synthesis with three EPS fractions from Cs-HK1 and proposed a modified protocol for Ag NP preparation. The properties and antibacterial activity of the Ag NPs were evaluated with respect to the molecular properties of the EPS fractions and experimental conditions.

## 2. Materials and Methods

### 2.1. Cs-HK1 Mycelial Fermentation and EPS Isolation

The Cs-HK1 fungus was isolated from wild *Cordyceps sinensis* and maintained in a mycelial culture as described previously [28]. For mycelial fermentation, the Cs-HK1 mycelium on a solid culture in a Petri dish was transferred into a liquid medium in an Erlenmeyer flask and incubated on a rotary shaker for 7 days. The mycelial culture broth was used as the inoculum for the subsequent mycelial fermentation, which was carried out in shake flasks at 20 °C for 7 days. The mycelial fermentation liquid was then centrifuged, and the liquid supernatant was collected for the isolation of EPS via ethanol precipitation. The ethanol precipitation was performed in a series of three steps with 40%, 60%, and 80% *v*/*v* ethanol, respectively, yielding three EPS fractions: EPS-1, EPS-2, and EPS-3. The EPS precipitates were separated from the liquid via centrifugation and freeze-dried as the final EPS products [29].

### 2.2. Characterization of EPS

The water solubility of EPS fractions was determined by dissolving a certain mass of EPS in deionized water and stirring vigorously overnight. The solution was then centrifuged at 6000 rpm for 10 min, and the undissolved precipitate was collected, dried, and weighed. The solubility of EPS fractions was the initial mass minus the undissolved mass per volume.

The total carbohydrate content in EPS fractions was determined with the Anthrone test using glucose as a calibration standard [30]. The EPS sample solution was prepared by dissolving 0.01 g EPS-1, EPS-2, or EPS-3 in 1 mL deionized water. After the solution was cooled to room temperature, the UV–Vis absorbance at 620 nm was recorded.

The total protein content in EPS fractions was determined with the Lowry method using bovine serum albumin (BSA) as a standard [31]. The EPS sample (0.01 g) was dissolved in deionized water to a final concentration of 10 mg mL^−1^. A Lowry Reagent Solution (1 mL) was added to 1 mL of the sample solution and kept at room temperature for 60 min, followed by the addition of 0.1 mL diluted Folin–Ciocalteu (FC) reagent solution (with deionized water at 1:1 volume ratio). The solution was kept at room temperature for another 45 min, and then absorbance at 750 nm was measured.

The molecular weight distribution of EPS fractions was analyzed via high-pressure gel permeation chromatography (HPGPC) with a Waters 1515 isocratic HPLC pump and a Waters 2414 refractive index detector. Two columns, Ultrahydrogel 120, 250 and Ultrahydrogel 2000 (both 7.8 mm × 300 mm dimensions from Waters Co., Milford, MA, USA), were used in a series. Deionized water was used as the mobile phase at 0.6 mL min^−1^ and 50 °C. The EPS sample was dissolved in deionized water at 0.4 mg mL^−1^ and filtered through a 0.45 μm membrane before injection. Dextran molecular weight standards ranging from 1.0 to 670 kDa (Sigma, St. Louis, MO, USA) were used for calibration. The Breeze V3.3 software was used for the computation of data.

The monosaccharide composition of EPS fractions was analyzed via HPLC after acid hydrolysis and a 1-phenyl-3-methyl-5-pyrazolone (PMP) reaction. In brief, 5 mg of sample was hydrolyzed with 2 mL of 2 M trifluoroacetic acid (TFA) at 110 °C for 4 h. The hydrolysate was then dried in a rotatory evaporator at 40 °C and redissolved in 2 mL of deionized water. The hydrolysate solution (450 μL) was mixed with 450 μL of a 0.5 M PMP solution in methanol and 450 μL of a 0.3 M NaOH solution, and the mixture was maintained at 70 °C for 30 min. The reaction was terminated by neutralization with 450 μL of 0.3 M HCl, and the product was partitioned with chloroform three times. The aqueous layer was collected, filtered through a 0.45 μm membrane, and applied to HPLC. The HPLC was performed with an Agilent ZORBAX ECLIPSE XDB-C18 column (150 mm × 4.6 mm) on an Agilent 1100 instrument at 25 °C with the mobile phase containing 15% potassium phosphate-buffered saline (0.05 M, pH 6.9) (solvent A) and 40% acetonitrile (solvent B). Absorbance at 250 nm was detected. Monosaccharide standards (Sigma) were used for the identification and quantification of the corresponding peaks.

### 2.3. Synthesis of Ag NPs

Ag NPs were synthesized as previously reported [24], with minor modifications. Silver nitrate AgNO_3_ (99.7%) was purchased from Sigma-Aldrich (ACS reagent, #209139). The EPS was dissolved in deionized water by stirring vigorously overnight and then centrifuged at 6000 rpm for 10 min to remove the undissolved part. The EPS solution was diluted to a final concentration of 0.5 g L^−1^. A silver nitrate (AgNO_3_) solution was prepared by dissolving 0.085 g of AgNO_3_ solids in 10 mL deionized water and diluting to 1, 5, and 10 mM, respectively. The total volume of the reaction mixture was set to 50 mL. The concentration of each solution referred to the final concentration of each component in the reaction mixture. The reaction was conducted in test tubes with constant stirring in the dark. The stirring and heating of the reaction solution were carried out with a magnetic stirrer hot plate; a relatively high stirring speed was applied for effective mixing. The UV–Vis absorbance of the solution mixture was recorded to monitor Ag NP formation as the reaction advanced.

The reaction between EPS and AgNO_3_ was carried out at three selected temperatures, 25, 70, and 100 °C, over a period of 5 h. The AgNO_3_ and reaction solutions were stored at room temperature (~25 °C) and in the dark before use.

### 2.4. Characterization of Ag NPs

The UV–Vis spectra of final solutions were measured from 300 to 600 nm on a HEWLETT Packard 8453 spectrophotometer against deionized water as blank.

The particle size of Ag NPs was measured via dynamic light scattering (DLS) at 25 °C with a scattering angle of 90° on a Malvern Zetasizer, model 3000 HSA.

The silver content of Ag NPs was determined via ICP-OES (Agilent Technologies 7000 Series, Santa Clara, CA, USA). The reacted solution was dialyzed with 3500 kDa MWCO bags (Sigma) against deionized water for two days at room temperature in the dark to remove unreacted silver ions [27]. The solid product was separated by freeze-drying to obtain EPS-coated Ag NPs, and 10 mg of the EPS-coated Ag NPs was dispersed in 12 mL of deionized water with stirring overnight. Then, the solution was filtered through a 0.45 μm membrane to remove the undissolved large particles and acid-digested before the ICP-OES analysis.

### 2.5. Antibacterial Assay

The in vitro antibacterial activity of EPS-coated Ag NPs was measured using *Escherichia coli* (Gram-negative) and *Staphylococcus aureus* (Gram-positive) in a 96-well microplate. The test was conducted according to the Clinical and Laboratory Standards Institute Guidelines [32] using its broth microdilution protocol. The bacteria were cultured on a Luria-Bertani [4] agar plate (Fluka Analytical, Sigma-Aldrich Co., St. Louis, MO, USA) overnight, and then 4–5 of the bacterial colonies were inoculated into 10 mL of LB broth and incubated at 37 °C for 4–6 h. The bacterial suspension was inoculated into a 96-well microplate (final concentration ~10^5^ CFU mL^−1^) containing 100 μL of serial dilutions of the tested samples. After incubation at 37 °C for 12 h, the absorbance at 600 nm was recorded to determine the minimum inhibitory concentration [5] with respect to the untreated control using a microplate reader.

## 3. Results and Discussion

### 3.1. Physical and Molecular Properties of EPS

Table 1 shows the water solubility, carbohydrate and protein contents, and monosaccharide composition of the EPS fractions. The solubility was the lowest for EPS-1, 0.711 g L^−1^, and the highest for EPS-3, over 60.0 g L^−1^. If the amount of EPS-1 added to water was higher than its solubility, the undissolved part would not contribute to the Ag NP synthesis since it would not be able to form a solution to react with AgNO_3_. The solubility of EPS-2 and EPS-3 was much higher than that of EPS-1. The EPS-1 fraction showed an extremely low carbohydrate content (Table 1), which could be partly attributed to the low solubility of EPS-1 since the analytical method could only measure soluble components in water. The increase in the total protein content from EPS-1 to EPS-2 and EPS-3 suggested that the proteins had lower molecular weights and were mainly precipitated at a higher concentration of ethanol.

All the three EPS fractions consisted of three major monosaccharides at different molar ratios, namely mannose (Man), glucose (Glc), and galactose (Gal). Appendix A shows the typical structure and NMR spectral data of EPS fractions, consisting of a heteroglycan main chain with side chains. According to the sequence of gradient precipitation, the first precipitated EPS-1 should have the highest MW, but it showed the lowest MW based on the HPGPC analysis. This result could be attributed to the low solubility of EPS-1 and its high MW constituents that were not well-dissolved in water and were, therefore, excluded by the 0.45 μm membrane before being injected into the HPGPC system.

### 3.2. Factors Affecting Ag NP Synthesis

#### 3.2.1. Effect of Reaction Time

According to the solubility results obtained above, the concentration of all the EPS fractions was fixed at 0.5 g L^−1^ for Ag NP synthesis. The first experiment was performed to monitor the formation of Ag NPs at a fixed concentration of AgNO_3_ and temperature in various time periods by the measurement of UV–Vis absorbance spectra. The appearance of an absorption peak due to the surface plasmon resonance (SPR) effect of NPs indicates NP formation in the solution [27]. The characteristic absorption peak for Ag NPs was found at the wavelength around 432 nm (Figure 1a). The absorption peak did not appear with 1–2 h reaction time and was obvious at 3 h, increasing with a further increase in the reaction time from 3 to 5 h. In general, the amount of the formed Ag NPs increases with the reaction time in the early period and approaches a plateau or maximum value at a later time. The reaction time for reaching the maximum amount is dependent on the different reducing materials [33,34,35,36]. The absorbance measurement suggested that the Ag NPs formed after 3 h of AgNO_3_ reacting with the EPS. The absorbance at 432 nm showed a rapid increase in the first 5 h of reaction but little change from 5 to 20 h (Figure 1b). Therefore, five hours was chosen in subsequent experiments.

#### 3.2.2. Effect of Reaction Temperature

The effect of reaction temperature on the formation of Ag NPs with EPS fractions was studied at 25, 70, and 100 °C, respectively, with 10 mM AgNO_3_ and 0.5 g L^−1^ EPS. As the reaction proceeded, the solution turned from colorless to yellowish brown, and its color became darker, resulting in an increase in absorbance at 432 nm (Figure 2), indicating the formation of more Ag NPs. For all the three EPS fractions (EPS-1, 2, and 3), the absorbance at 432 nm increased with a higher reaction temperature, implying that a higher temperature accelerated the formation of Ag NPs. At a given temperature, the absorbance at 432 nm initially increased with time until reaching a plateau, which meant nearly all the Ag(I) ions were consumed, and no more Ag NPs formed. The higher rate of Ag NP formation is consistent with the trend found in previous studies [33,37], mainly because a higher temperature confers a higher kinetic energy to the molecules to accelerate the reaction rate. At the highest temperature of 100 °C, the curve reached its plateau much quicker than at lower temperatures. Since the formation of Ag NPs was faster at 100 °C, the reaction temperature of 100 °C was chosen for subsequent experiments.

#### 3.2.3. Effect of AgNO_3_ Concentration

The effect of AgNO_3_ concentration on Ag NP synthesis was evaluated at three AgNO_3_ concentrations 1 mM, 5 mM, and 10 mM with the other conditions fixed. Figure 3 shows the absorbance at 432 nm of solution mixture recorded over 5 h of reaction. In most cases, the absorbance initially increased with time and then reached a plateau. For EPS-1 and EPS-2, the absorbance at 5 mM AgNO_3_ was the highest, while for EPS-3, the absorbance was similar for 5 mM and 10 mM AgNO_3_ and much lower for 1 mM AgNO_3_. A higher concentration of AgNO_3_ did not necessarily result in the formation of more Ag NPs, which was consistent with the trend from a previous study [27]. This phenomenon is most probably attributed to the excess of more silver ions over the active sites on the biomolecules available for reduction [33]. As the active sites on biomolecules were fully occupied by silver ions, the extra silver ions could not bind to the biomolecules, nor could they be reduced to atomic silver, so that a further increase in the AgNO_3_ concentration would not promote the formation of Ag NPs.

To identify the optimal conditions for Ag NP synthesis, the particle size of Ag NPs was also measured using DLS (Appendix A). With EPS-1 and EPS-2, the average sizes of Ag NPs synthesized with 5 mM AgNO_3_ were the smallest, below 175 nm and 215 nm, respectively. With EPS-3, the average size with 10 mM AgNO_3_ was the smallest, ~240 nm, and the average particle size at 1 mM AgNO_3_ was more than six times as large. Moreover, the large particle size was accompanied by large variance, indicating the poor uniformity of Ag NPs. Therefore, to obtain a small and uniform particle size of Ag NPs, the optimal AgNO_3_ concentration was 5 mM for EPS-1 and EPS-2, and 10 mM for EPS-3.

The optimal reaction time was determined by the sample with high UV–Visible absorbance and small particle size in the meantime. Considering both criteria, the optimal reaction time was 3 h for EPS-1, 5 h for EPS-2, and 1 h for EPS-3. The optimal conditions of Ag NP synthesis are summarized in Table 2.

#### 3.2.4. Mechanism of Ag NP Synthesis with EPS

Figure 4 depicts the possible reaction mechanism for the formation and stabilization of Ag NPs with EPS fractions. The whole process involves three major steps as follows: The first step is the reduction of Ag+ ions to Ag atoms by the EPS, which has significant reducing activity, as previously reported [27]. In the next step, the Ag atoms agglomerate to form Ag NP crystals. In the third step, the Ag NPs are capped or coated by a layer of EPS and may also be adsorbed to the EPS networks, thus being stabilized in the aqueous solution. The formation of aggregated gel networks in an aqueous solution is characteristic of water-soluble polysaccharides.

### 3.3. Antibacterial Activities of Ag NPs

The prepared Ag NPs were dialyzed to remove the unreacted Ag(I) ions, if any, in the solution and to avoid interference with the antibacterial activity assay. The average particle size of Ag NPs increased significantly after dialysis due to the removal of smaller particles (Appendix A). Moreover, coagulation was also observed during the dialysis process, contributing to the increase in particle size. The hydro-diameter data were consistent with the molecular weight trend for EPS-1, EPS-2, and EPS-3 dissolved in water, suggesting the correlation of the particle size of Ag NPs to the hydrocolloid size of the EPS.

The atomic silver content in EPS-coated Ag NPs after dialysis was measured via ICP-OES. The solution was filtered through a 0.45 μm membrane, and the average silver content values were 2.52%, 1.77%, and 3.23% (*w*/*w*) for EPS-1 Ag NPs, EPS-2 Ag NPs, and EPS-3 Ag NPs, respectively (Table 3). After filtration, large particles were removed, and the solution color became lighter and more transparent. As estimated from the ICP-OES test, about 50% of silver was removed in the filtration process.

Table 4 shows the results of the antibacterial activity assay. All the Ag NPs formed with EPS-1, 2, and 3 showed the ability to inhibit the bacterial growth, among which the Ag NPs formed with EPS-1 exhibited the strongest inhibiting effect. EPS-1 Ag NPs completely inhibited the growth of *E. coli* (Gram-negative) at 0.2 mg/mL and *S. aureus* (Gram-positive) at 0.075 mg/mL. A stronger inhibiting effect on Gram-positive bacteria was observed for EPS-1 Ag NPs, which was consistent with previously reported results [27]. The higher antibacterial activity of EPS-1 Ag NPs may partially be attributed to their smaller particle size than those of the EPS-2 Ag NPs and EPS-3 Ag NPs, as smaller Ag NPs can more intimately interact with the bacteria. EPS-1, 2, and 3 did not inhibit any of the bacteria without the existence of Ag NPs. In summary, the EPS-1 Ag NPs exhibited the highest inhibiting effects for both Gram+ and Gram- bacteria among the three fractions of EPS-coated Ag NPs.

## 4. Conclusions

In this study, EPS-coated Ag NPs were synthesized by AgNO_3_ and three fractions of exopolysaccharide (EPS) produced by a medicinal fungus known as Cs-HK1. The effect of various factors on Ag NP synthesis, including reaction time period, reaction temperature, and reagent concentration, were studied, and the optimal conditions for Ag NP synthesis were investigated. The overall synthesis procedure only involved water without the addition of any organic solvents. Moreover, the Ag NP preparation did not require any harsh experimental conditions. Therefore, this Ag NP synthesis method is green, facile, and convenient. Moreover, the EPS-coated Ag NPs showed strong antibacterial activity against *E. coli* and *S. aureus*. The most significant bacterial inhibiting effect was found with EPS-1 Ag NPs. Further studies may focus on the specific components and structures of the EPS in order to gain a better understanding of synthetic pathways and to seek suitable applications of EPS-coated Ag NPs.

## Figures and Tables

**Figure 1 materials-15-05620-f001:**
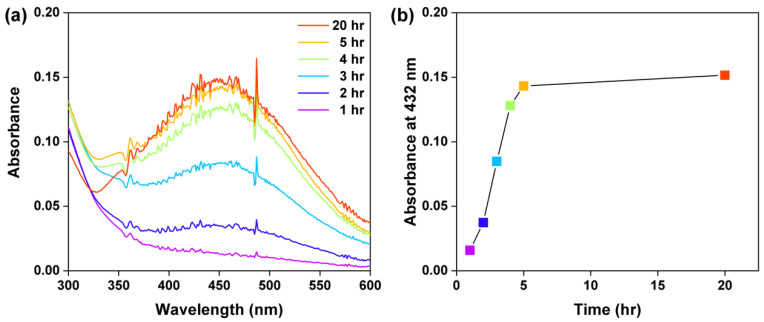
(**a**) UV–Vis spectra and (**b**) UV–Vis absorbance at 432 nm (the characteristic peak) of Ag NPs synthesized at 25 °C with various reaction times with 0.5 mg mL^−1^ EPS-1 and 10 mM AgNO_3_ solution.

**Figure 2 materials-15-05620-f002:**
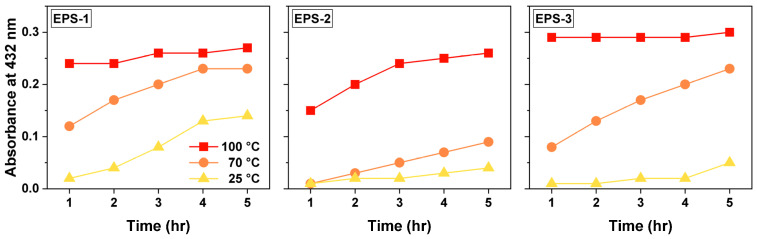
UV–Vis light absorbance at 432 nm of AgNO_3_ and EPS solution mixture at 25, 70, and 100 °C, respectively. Each reaction mixture contained 10 mM AgNO_3_ and 0.5 g L^−1^ EPS.

**Figure 3 materials-15-05620-f003:**
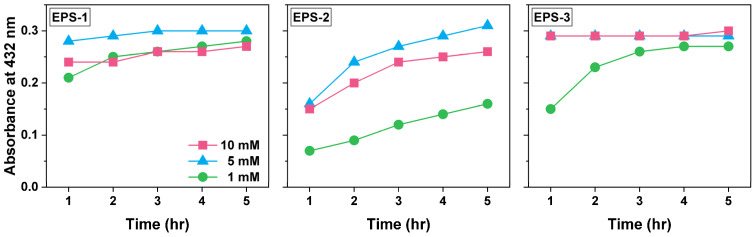
UV–Visible light absorbance at 432 nm of AgNO_3_ and EPS solution mixture as a function of reaction time with the AgNO_3_ concentration of 1, 5, and 10 mM. Each reaction mixture contained 0.5 g L^−1^ EPS, and the reactions were conducted at 100 °C.

**Figure 4 materials-15-05620-f004:**
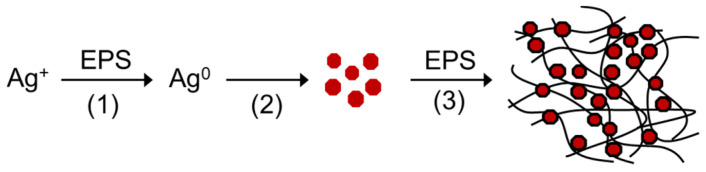
Mechanism for formation and stabilization of AgNO_3_ with EPS: (**1**) reduction of Ag+ to Ag atoms; (**2**) crystallization of Ag atoms to Ag NPs; (**3**) capping of Ag NP by EPS and adsorption of Ag NPs to EPS networks.

**Table 1 materials-15-05620-t001:** Water solubility and molecular properties of three EPS fractions.

EPS	Solubility (g L^−1^)	Total Carbohydrate (wt%)	Total Protein (wt%)	MW (Da)	Man:Glc:Gal Molar Ratio
EPS-1	0.711 ± 0.08	13.8 ± 1.9	12.2 ± 0.3	6.498 × 10^5^	2.8:7.9:1
EPS-2	17.4 ± 0.20	67.2 ± 1.7	40.1 ± 0.3	3.860 × 10^8^	16:1:7
EPS-3	>60.0	39.5 ± 1.8	43.6 ± 1.8	9.221 × 10^6^	11.4:1:10.1

*Note:* Supplementary data provide the original analytical data: Appendix A. HPGPC spectra; Appendix A. Data for the molecular weight distributions of three EPS fractions; Appendix A. HPLC analysis of monosaccharide standards and compositions of EPS fractions.

**Table 2 materials-15-05620-t002:** Summary of the optimal conditions for the synthesis of Ag NPs with three fractions of EPS.

EPS	EPS-1	EPS-2	EPS-3
Time (h)	3	5	1
Temperature (°C)	100	100	100
EPS conc. (g L^−1^)	0.5	0.5	0.5
AgNO_3_ conc. (mM)	5	5	10
UV–Vis absorbance at 432 nm	0.3	0.31	0.29
Ave. particle size (nm)	158.9 ± 5.1	213.7 ± 9.2	170.1 ± 5.6

**Table 3 materials-15-05620-t003:** Silver content of EPS-coated Ag NPs after filtration by 0.45 μm membrane.

EPS-Coated Ag NPs	Average Ag Content (wt%)
EPS-1 Ag NPs	2.52 ± 0.03
EPS-2 Ag NPs	1.77 ± 0.01
EPS-3 Ag NPs	3.23 ± 0.12

**Table 4 materials-15-05620-t004:** The antibacterial activity of Cs-HK1 EPS fractions and corresponding EPS-coated Ag NPs after incubation at 37 °C for 12 h.

Sample	*E. coli* MIC ^a^ (mg mL^−1^)		*S. aureus* MIC (mg mL^−1^)	
	EPS	Ag NPs	EPS	Ag NPs
EPS-1 ^b^	>0.2	0.2 ± 0.01	>0.2	0.075 ± 0.01
EPS-2	>0.8	0.8 ± 0.01	>0.8	0.8 ± 0.02
EPS-3	>0.8	0.6 ± 0.01	>0.8	>0.8

^a^ Minimum inhibitory concentration of samples. ^b^ The maximum concentration of EPS-1 applied was 0.2 mg mL^−1^ due to poor solubility.

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
