# Peer review of "Synthetic Conditions, Physical Properties, and Antibacterial Activities of Silver Nanoparticles with Exopolysaccharides of a Medicinal Fungus"

_materials, 2022, doi:10.3390/ma15165620_

Round 1

Reviewer 1 Report

Thanks to the authors for having submitted their manuscript. 

The work is fine and nicely presented, but it is not novel enough. There are indeed similar papers on the same topic (synthesis of silvernanoparticles with EPS) and I don't think this paper should be published in a journal with this impact factor.

I would like to suggest some improvements on the paper: chemical characterisation of EPS (NMR, FT-IR) and silver nanoparticles (zeta-potential and SEM/TEM). Could you please specify at what stirring rate where the reactions kept during the synthesis of AgNPs? Why the AgNPs were not digested in acid before running ICP-OES?

Author Response

Although we have previously studied on the use of EPS for Ag NPs synthesis as cited in the Ms, the present study provided new and useful information for better understanding of the major conditions and factors in Ag NPs synthesis with EPS. Even though the topic is not wholly new, it still provides practical and useful information for green synthesis of Ag NPs synthesis with natural polysaccharides.

Response to specific comments/questions.

  1. Chemical characterization of EPS and Ag NPs

Table 1 provides the major molecular properties of EPS fractions available for this study. The molecular weight range presented in Table 1 was the major variable but not the molecular structure. Characterize of their structures was not an objective of the present study.

  1. Stirring rate

As added in the revised Ms, “The stirring and heating of reaction solution was provided with a magnetic stirrer hot plate; relatively stirring speed was applied for effective mixing.” The exact stirring speed was not included as a variable.

  1. ICP-OES determination of Ag contents

As stated in the Ms (line 141): “Then the solution was filtered through 0.45 mm membrane to remove the undissolved large particles and acid digested before the ICP-OES analysis.”

Reviewer 2 Report

This is a well prepared paper reporting the synthesis of Ag NPs using milde conditions, their physical characterization and antibacterial activity. Although the topic is not new, I acknowledge the fact that the use of natural molecules as reducing agents as well as capping agents, has  interest for related research in the field.

The idea is maintained through the text and is rather clear. The conclusions are supported by experimentally obtained results.

I am of the opinion that the paper can be accepted for publication.

Author Response

Not needed.

Reviewer 3 Report

The manuscript is exciting, but some modifications are necessary.

1. Supplementary material has not been added to the manuscript.

2. in the abstract, the variation and the result of the antimicrobial activity should be more explicit

3. I did not understand how the experiments were performed with light. The reactions were conducted in test tubes with constant stirring in the dark, except for the experiments studying the effect of light.

4. The results of the effect of time, concentration, and temperature should be better explored, including a comparison with the literature and an explanation for their impact.

5. Images of the antimicrobial activity tests should be added, and a closing of the text showing which was the best material with the best activity and the reason.

6. The same goes for the conclusion.

Author Response

  1. Added the missing supplementary materials.
  2. Added more description of antimicrobial activity test results to abstract part.
  3. We tested the effect of light exposure on the AgNP synthesis in our preliminary experiments but not presented the results. Therefore, we deleted all related statements in the revised Ms: line 17 in Abstract, line 124 in Materials and Methods, and line 288 in Conclusions.
  4. Added a comparison to literature to explain the effects of each conditions: line 190-193 for reaction time, line 209-211 for temperature, and line 226-231 for Ag concentration.
  5. More explicit description of antimicrobial activity results and discussion were added to the text. A summary of the material with the best antimicrobial activity was also added to the text. Unfortunately, due to the limited amount of EPS coated Ag NPs, the experiments were conducted in 96-well plates to save materials and we only used them for quantitative results. Due to the limited area of each well in the plates, we could not provide good quality images to show the antimicrobial activity. We may try to scale up the experiments and to test antimicrobial effects in petri dish in the future work.
  6. One statement of antimicrobial activity test results was added to the conclusion part.

Reviewer 4 Report

The article "Synthesis conditions, physical properties and antibacterial activities of silver nanoparticles with exopolysaccharides of a medicinal fungus" is very interesting and written well. They have evaluated the synthesis conditions, physical properties and potential antibacterial applications of silver nano particles coated with exopolysaccharides produced by a medicinal fungus Cs-HK1. Introduction was written well and provided a nice overview. Also results were presented in a good format. However they have not provided much discussion and comparison with similar studies. The exopolysaccharides structure have to be included in this article. In addition, they have to provide a figure which shows how the exopolysaccharides were attached to the silver nano particles. Hence I recommend the article for a publication after a major review. 

Author Response

  1. Added discussion and comparison with published studies to explain the effects of each conditions: line 190-193 for reaction time, line 209-211 for temperature, and line 226-231 for Ag concentration.
  1. Chemical characterization of EPS and Ag NPs: Table 1 provides the major molecular properties of EPS fractions available for this study. The molecular weight range as shown in Table 1 was a major variable but not the molecular structure. As responded to Reviewer #1, characterization of the EPS structures was beyond the scope of present study.
  1. Added a new section “3.2.4. Mechanism of Ag NP synthesis with EPS” and Figure 4, to illustrate the attachment of Ag NPs to EPS networks as well as the overall reaction mechanism for the formation and stabilization of Ag NPs with EPS.

Round 2

Reviewer 4 Report

The article "Synthesis conditions, physical properties and antibacterial activities of silver nanoparticles with exopolysaccharides of a medicinal fungus" is improved compared to the previous version. However they have not provided any structure of the exopolysaccharides(EPS), which is really important for these kind of studies. It is is not possible to use those results without knowing the structure of the exopolysaccharides and they can not reproducible to others. They have to investigate the structure of the EPS using NMR, mass spectroscopy or similar methods, before claiming its antibacterial activities. Hence I recommend the article to publish after a major revision.  

Author Response

Added "Figure S8 Structure analysis and characterization of EPS-3" in Supplementary data.

In revised Ms (line 168): "Figure S8 shows the typical structure and NMR spectral data of EPS fractions, consisting of a heteroglycan main chain with side chains."